# Association between Low House Cleaning Frequency, Cough and Risk of Miscarriage: A Case Control Study in China

**DOI:** 10.3390/ijerph18105304

**Published:** 2021-05-17

**Authors:** Fumei Gao, Xiangrui Meng, Qiuxiang Zhang, Min Fu, Yumeng Ren, Jianying Hu, Huan Shen, Kun Tang

**Affiliations:** 1Reproductive Center of Peking University Peoples’ Hospital, Peking University, Beijing 100044, China; fumei_gao@pku.edu.cn (F.G.); 13701294162@163.com (Q.Z.); kinki836@souhu.com (M.F.); 15232236997@163.com (Y.R.); 2Vanke School of Public Health, Tsinghua University, Haidian District, Beijing 100084, China; harrymengpku@outlook.com (X.M.); tangk@mail.tsinghua.edu.cn (K.T.); 3Laboratory for Earth Surface Processes, College of Urban and Environmental Sciences, Peking University, Beijing 100871, China; hujy@urban.pku.edu.cn

**Keywords:** house cleaning frequency, cough, risk of miscarriage

## Abstract

This study investigated the association between house cleaning frequency and the risk of miscarriage in a case control sample of Chinese population. We recruited 59 pregnant women with clinical pregnancy loss as cases and 122 women who chose to conduct induced abortion as controls. All participants were aged 20~40 years and completed a questionnaire of lifestyle exposure with a trained nurse. The effect of frequency of cleaning up on risk of miscarriage was estimated using multivariable logistic regressions, adjusting for potential confounders. In the present study, it was shown that house cleaning of less than twice per week was significantly associated with cough during day or night with odds ratio (OR) of 2.97 (95% CI: 1.36~6.75, *p* = 0.007), and cough during day or night was significantly associated with risk of miscarriage with OR of 2.69 (95% CI: 1.22~6.02, *p* = 0.014). Thus, house cleaning of less than twice per week was statistically significantly associated with miscarriage with OR of 3.05 (95% CI: 1.51~6.31, *p* = 0.002). We found that females who have their house cleaned less than twice per week are at elevated risk for miscarriage. Therefore, the home of pregnant woman should be cleaned at least twice per week in order to avoid miscarriage.

## 1. Introduction

Early pregnancy loss or first-trimester miscarriage is one of the most common complications of female reproduction, with an incidence of 17–22% of all recognized pregnancies [1]. The true rate of pregnancy loss is difficult to determine, and some authors have suggested 20–40% of all losses may occur before clinical detection [2,3]. Moreover, the incidence of the disease has been increasing worldwide in recent decades [4]. Although many factors affecting early pregnancy loss have been identified in recent years, the causes for approximately 40% of early pregnancy loss remains unclear [5]. Few studies have evaluated the effect of lifestyle exposure on risk of miscarriage.

Since pregnant women might spend more time staying at home, indoor pollutants at home would impact the risk of early pregnancy loss. It has been suggested that exposure to common indoor environmental allergens could induce the occurrence of asthma, the main clinical manifestation of which is cough. Maternal asthma is associated with an increased risk of a spontaneous abortion due to the dysfunctional immune response. In addition, household dust was important indoor environmental pollutants that may have notable health effects on humans due to chronic exposure. The household dust, as a medium, contained a lot of chemicals such as particulate matter (PM), endocrine disruptor chemicals (EDCs), NO_2_, SO_2_, etc. Among these, EDCs, similar in chemical structure to hormones, interfere their normal functioning; thus, an exposure to their action (especially in the prenatal and early life) may lead to increased risks of cancers, obesity, other metabolic and fertility disorders, preterm birth, miscarriage, developmental disorders, and epigenetic changes in subsequent generations [6,7]. The reproductive effect of EDCs has recently become of great interest. As home environment is one of the major sources of household pollution for pregnant women, health organizations emphasize the need of implementing lifestyle changes to protect human health [8]. Therefore, we design a questionnaire to explore the association between lifestyle, especially the house cleaning frequency, and miscarriage, and proposed to provide the practical and feasible tips of staying at home for the women who have conceived.

## 2. Material and Methods

### 2.1. Setting and Sample

Subjects with clinical pregnancy loss were recruited as cases for this study from the Department of Obstetrics and Gynecology of Peking University People’s Hospital in Beijing, China, from May 2018 to December 2019. According to the diagnosis of clinical pregnancy loss, when women meet any of the criteria: (1) the mean sac diameter of their gestation sac is greater than 20 mm without a visible embryo, (2) the length of their embryo is more than 6 mm with invisible activity, or (3) repeat trans-vaginal ultrasound showing an absence of cardiac activity in embryos or fetus, the women were confirmed as clinical pregnancy loss. Trans-vaginal ultrasound examination was used to check the development condition of embryos by experienced doctors. During the same time, patients with a fetal bud body with a heartbeat under the trans-vaginal ultrasound examination who want to conduct induced abortion, were invited to participate in this study as controls. The control and case population are from the same population in the outpatient during the same time. Gestation period of less than 20 weeks and maternal age of 20~40 years were compulsory to be considered as eligible participants for both case and control groups. On the basis of a physical examination and self-report questionnaire, subjects with recurrent miscarriages, history of infertility, reproductive tract abnormalities, gynecological inflammation, fever during early pregnancy, or husband in bad health were excluded from the study. This study was approved by the Ethics Committee of Peking University People’s Hospital (2011-33). All participants were informed of the design and signed an informed consent.

### 2.2. Measurements

Each subject completed a face-to-face questionnaire which included information regarding demographics (age, body mass index (BMI), nationality, and week of gestation), current smoking or drinking status, household income, lifestyle factors under the guidance of a trained investigator.

House cleaning behavior of participants were asked by the questionnaire with the following options: no less than twice per week, one per week or less than twice per week. In any subsequent analysis, participants cleaned their house less than twice per week were defined as the exposed group, while participants cleaned their house no less than twice per week were defined as the non-exposed group. Risk of miscarriage was modelled for the exposed group compared with the non-exposed participants. Age of participants were calculated from their dates of birth. Weights and heights of participants were reported by patients. In the questionnaire, participants were asked whether they usually cough during day or night with the following options: no, yes and last for less than 3 months, yes and last for no less than 3 months. Smoking were required in the questionnaire as ‘How often do you smoke now?’ with 4 options: do not smoke, seldomly smoke, smoke on most days or smoke every day. Similarly, participants were also asked how often do they drink at the time of baseline assessment with five options: do not drink, drink only occasionally, drink on specific months, drink every month by less than once per week, drink at least once per week. Participants were asked how often they workout or do physical exercise in their spare time in the past year, with 5 options: never or seldomly, 1–3 times per month, 1–2 times per week, 3–5 times per week, every day or nearly every day. Income of the patient were asked with 3 options (per month): <7000 yuan, 7000~10,000 yuan, >10,000 yuan. The highest qualification participants attained were asked by questionnaire with the following 6 options: no formal education, primary school, middle school, high school, college, university or above.

### 2.3. Statistical Analysis

The difference in house cleaning behavior between cases and controls were first tested by logistic regression model adjusting for age and BMI. In addition, the association between low house cleaning frequency and cough, between cough and risk of miscarriage were also tested by logistic regression models adjusting for age and BMI. In order to further explore the association between exposure to low house cleaning frequency and risk of miscarriage, other logistic regression models were further fitted. In model 1, age and BMI were adjusted. In model 2, cough was further adjusted with age and BMI to explore whether the association between low house cleaning frequency and risk of miscarriage executes via cough. Additionally, in model 3, we adjusted smoking status, drinking status, physical activity frequency, household income and highest education with age, BMI and cough.

Moreover, we tested the difference in house cleaning frequency between cases and controls stratified by cough or not, age or BMI. The study sample was split according to whether they cough or not, their age (<35 years, or ≥35 years), or their BMI (<25 kg/m^2^ or ≥25 kg/m^2^). Two logistic models were fitted in each of the six stratified population. Age, BMI and cough were adjusted in model 1 if applicable, and smoking status, drinking status, physical activity frequency, household income and highest education were further adjusted in model 2.

## 3. Results

In total, there were 59 miscarriage patients (case) and 122 induced abortion patients (control). Mean age for them were 33.34 ± 5.26 and 32.34 ± 6.68, respectively. Mean BMI for them were 21.84 ± 2.88 and 21.00 ± 3.02, respectively. There were no significant differences between case and control groups in terms of smoking status, drinking status, household income, highest education and occupation (Table 1). In the case populations, 17 reported a usual cough during day or night (42 no symptoms of cough), while 14 among the 122 control patients showed a significant *p* value. In addition, a lower house cleaning frequency (less than twice per week) was also demonstrated in the case group compared to the control patients (*p* = 0.005). However, the samples size was too small to model a dose–response relationship.

In our inter-relationship test for house cleaning frequency, cough and risk for miscarriage, house cleaning of less than twice per week was significantly associated with cough during day or night with OR of 2.97 (95% CI: 1.36~6.75, *p* = 0.007). Cough during day or night was significantly associated with risk of miscarriage with OR of 2.69 (95% CI: 1.22~6.02, *p* = 0.014). House cleaning of less than twice per week was statistically significantly associated with miscarriage with OR of 3.05 (95% CI: 1.51~6.31, *p* = 0.002).

When cough got adjusted in model, low house cleaning frequency was still statistically significantly associated with risk of miscarriage but its estimated effect attenuated (OR = 2.70, 95% CI: 1.31~5.66, *p* = 0.008) (Table 2). In addition, cough reach a nominal significance in our model (OR = 2.19, 95% CI: 0.96~5.03, *p* = 0.061). When other confounders were further adjusted, the association between low house cleaning frequency and risk of miscarriage was still significant (OR = 2.82, 95% CI: 1.32~6.18, *p* = 0.008), while cough held at a statistically suggestive level (OR = 2.28, 95% CI: 0.97~5.41, *p* = 0.058). Age, BMI, smoking status, drinking status, physical activity frequency, household income or highest education were not associated with the risk for miscarriage (Table 2).

In our stratified analyses, we observed significant association in participants without symptom of cough (OR = 2.90, 95% CI: 1.24~6.98), participants < 35 years old (OR = 3.74, 95% CI: 1.25~12.29) and participants with BMI < 25 kg/m^2^ (OR = 3.88, 95% CI: 1.65~9.54) (Table 3). Significant associations were not identified for participants with symptom of cough, ≥35 years old or with BMI ≥ 25 kg/m^2^. In the analyses, we were unable to show any significant moderation effect of cough, age or BMI on the association between house cleansing frequency and risk of miscarriage.

## 4. Discussion

In the present study, we explored the association between lifestyle and miscarriage. The lifestyle questionnaire included age, body mass index (BMI), smoking, drinking, cough, physical activity, household income, education and occupation. We found that in the miscarriage (case) groups, patients showed more symptoms of cough and lower house cleaning frequency, compared to the patients of control. Then, we further analyzed the inter-relationship during the three elements. It was found that house cleaning of less than twice per week was significantly associated with cough during day or night (OR = 2.97, 95% CI: 1.36~6.75, *p* = 0.007), and cough during day or night was significantly associated with risk of miscarriage with OR of 2.69 (95% CI: 1.22~6.02, *p* = 0.014). Moreover, it was found that house cleaning of less than twice per week was statistically significantly associated with miscarriage with a high OR of 3.05 (95% CI: 1.51~6.31, *p* = 0.002). Otherwise, we verify the association of house cleaning frequency and miscarriage using various models and showed an increased risk of miscarriage in low house cleaning frequency patients in the different models. Moreover, in the stratified analyses, this trend was obvious in participants without symptom of cough, participants < 35 years old, and participants with BMI < 25 kg/m^2^.

In the present study, we found that house cleaning of less than twice per week was significantly associated with cough during day or night with OR of 2.97 (95% CI: 1.36~6.75, *p* = 0.007). Cough, as the sole or predominant symptom of asthma, reflected the inflammatory or immune response of the airway [9]. A large body of evidence suggested that exposure to common indoor environmental allergens, including the house dust mite allergens, was an important risk factor for asthma development. Thus, several studies have focused on the methods of exposure reduction to house dust mite allergens. It was found that the cleaning intervention significantly lowered dust loading in all households and reduced the incidence of asthma in low-income, urban home environments [10,11]. The results were similar to the findings in our study. The higher the frequency of cleaning up (more than twice per week), the less incidence of cough (OR 2.97, 95% CI: 1.36~6.75, *p* = 0.007). In this point, we conclude that the increased frequency of cleaning up offer practical, effective means of decreasing the incidence of cough during day or night by reducing the exposure to house dust allergen levels.

In addition, it was found that cough during day or night was significantly associated with risk of miscarriage with OR of 2.69 (95% CI: 1.22~6.02, *p* = 0.014). A cohort of pregnancies from asthmatic (*n* = 15,107) and non-asthmatic (*n* = 34,331) women demonstrated that maternal asthma was associated with an increased risk of a spontaneous abortion (OR = 1.41; 95% CI: 1.33–1.49) [12]. One proposed mechanism to explain the increased risk of a spontaneous abortion in women with asthma was dysfunctional immune response. The nonatopic phenotype of asthma, classified as T2-low asthma/non-T2 Type 1 (T1) asthma, accounts for about 50% of asthma [13]. This type of asthma was characterized by biomarkers of neutrophil recruitment and innate immune response dysregulation. The proinflammatory T2-low asthma endotype has similar biomarker characteristics to the fetal-maternal interface during implantation. Prior to implantation and in the peri-implantation period the fetal–maternal interface is also characterized by immune deviation to a proinflammatory and T2-low endotype [14]. The same mechanism underlying the T2-low asthma and miscarriage explained the high incidence of miscarriage in asthmatic women. This maybe also the reason for the association between cough and miscarriage in our study. However, there was a big difference between the ORs in our study and the literature. It may be due to the difference of the populations. In our study, the population just had a clinical syndrome of cough, but not the diagnosis of asthma, though cough was a predominant symptom of asthma. While the populations in the literature were diagnosed as asthma. Thus the dysfunctional immune response was more serious in asthma population with a bigger OR.

Furthermore, the results in the study also showed that house cleaning of less than twice per week was statistically significantly associated with miscarriage with OR of 3.05 (95% CI: 1.51~6.31, *p* = 0.002). As we know, the more frequency of cleaning up, the less household dust in the room. Most people spend nearly 90% of their time indoors [15] so there are much higher opportunities for people to be exposed to contaminants in the household dust than in the outdoor environment. The household dust, as a medium, contained a lot of chemicals such as particulate matter (PM), EDCs, NO_2_, SO_2_, etc. There was some evidence that showed the relationship between these household chemicals’ exposure and miscarriage [16,17], which may contribute to the association between cleaning up frequency and miscarriage in our study. Thus, in order to reduce the risk of miscarriage, keeping a good cleaning up habit (more than twice per week) was a simple and feasible intervention for women who have conceived.

In the stratified analyses, we found that the association of lower house cleaning frequency and high risk of miscarriage in participants without symptom of cough, participants < 35 years old, and participants with BMI < 25 kg/m^2^. This may due to the low statistical power caused by low sample size in these three groups (N = 35, N = 89, N = 39, respectively). In addition, the house cleaning may be just a potential and weak influencing factor for the risk of miscarriage. In patients of advanced age, obesity and with the symptom of cough, the effect of house cleaning would not be obvious. This is why the people do not pay much attention to the habit of house cleaning.

Still, important limitations need to be considered when interpreting the findings. Our study sample is of limited size, thus we might not have sufficient statistical power for specific subpopulations (e.g., participants without symptom of cough, ≥35 years old or ≥25 kg/m^2^). However, the significant association between house cleaning frequency and risk for miscarriage was consistently observed in different models and different populations with OR of greater than 2. In addition, we observed a similar significant association between house cleaning frequency and risk for miscarriage in an independent sample (our previous study consisted of 152 healthy pregnancy patients and 107 patients of miscarriage, the results also showed that less cleaning up frequency was associated with high risk of miscarriage with OR of 3.50, 95% CI: 1.60–7.64, *p* = 0.00). These findings support the robustness of our findings. In addition, we did not have information on the levels of chemical pollutants such as EDCs, PM2.5, or PM10 in the household dust or the biomarker in the human serum that would be helpful to delve into potential biological mechanisms underlying the association between cleaning up frequency and pregnancy loss. Future studies with larger sample sizes, with collection of household dust samples and blood biomarkers, or even local biomarkers, would be help in the unraveling the true mechanism underlying the association between house cleaning and risk for miscarriage.

## 5. Conclusions

In conclusion, we observed that females who have their house cleaned less than twice per week are of elevated risk for miscarriage. The associations are also significant for participants younger than 35 years old and with BMI of <25 kg/m^2^. Therefore, homes of pregnant woman should be cleaned at least twice per week in order to avoid miscarriage.

## Figures and Tables

**Table 1 ijerph-18-05304-t001:** Characteristics of participants by their house cleaning frequencies.

	Case (N = 59)	Control (N = 122)	*p* Value
Age	33.34 (5.26)	32.34 (6.68)	0.27
Body mass index	21.84 (2.88)	21.00 (3.02)	0.08
Cough ^a^	17/59	14/122	0.004
**Smoking (%)**			0.70
Non-smoker	54 (91.50)	107 (87.7)	
Smoker	5 (8.50)	15 (12.30)	
**Drinking (%)**			0.30
Non-drinker	27(45.80)	46 (37.70)	
Drinker	32 (54.20)	76 (62.30)	
**Physical activity**			0.24
Less than once per week	38 (64.40)	89 (73.00)	
No less than once per week	21 (35.60)	33 (23.00)	
**Household income (per month)**			0.71
<7000 yuan	22 (37.3)	53 (43.40)	
7000~10,000 yuan	16 (27.10)	28 (23.00)	
>10,000 yuan	21 (35.60)	41 (33.60)	
**Highest education**			0.59
College or below	17 (28.80)	40 (32.80)	
University or above	42 (71.20)	82 (67.20)	
**Occupation**			0.07
Worker	1 (1.70)	6 (4.90)	
Manager	18 (30.5)	45 (36.90)	
Technical staff	29 (49.2)	43 (35.20)	
Homework or unemployment	7 (11.90)	7 (5.70)	
Other	4 (6.80)	21 (17.20)	
**Cleaning house frequency**			0.00
<1 per week	4 (6.78)	6 (4.92)	
1 per week	26 (44.07)	30 (24.59)	
2 per week	27 (45.76)	81 (66.39)	
>2 per week	2 (3.39)	5 (4.10)	

^a^. Participants reported themselves as always cough during day or night.

**Table 2 ijerph-18-05304-t002:** Association between house cleaning frequency and miscarriage adjusting for common confounders.

	Model 1 ^a^	Model 2 ^b^	Model 3 ^c^
House cleaning (less than twice per week)	3.05 (1.51, 6.31)	2.70 (1.31, 5.66)	2.82 (1.32, 6.18)
Age	1.04 (0.98, 1.11)	1.04 (0.98, 1.11)	1.02 (0.96, 1.09)
BMI	1.08 (0.95, 1.22)	1.07 (0.94, 1.21)	1.07 (0.94, 1.22)
Cough during day or night		2.19 (0.96, 5.03)	2.28 (0.97, 5.41)
Current Smoker			0.31 (0.05, 1.27)
Current Drinker			0.59 (0.28, 1.22)
Physical activity ≥ 1 per week			1.59 (0.74, 3.41)
Household income 7000~10,000 yuan (vs. <7000 yuan) per month			1.42 (0.57, 3.51)
Household income > 10,000 yuan (vs. <7000 yuan) per month			1.47 (0.58, 3.79)
Qualification of University of above			0.84 (0.35, 2.00)

^a^. Adjusted for age and BMI. ^b^. Adjusted for age, BMI, cough frequency. ^c^. Adjusted for age, BMI, cough frequency, physical activity, smoking, drinking, household income and highest education.

**Table 3 ijerph-18-05304-t003:** Odds ratio for self-reported cough, age group or body mass index of those with miscarriage with a house cleansing of lower than twice per week.

	Case	Control	OR (Model 1) ^c^	OR (Model 2) ^d^	*p* ^e^
	No. of Freq < 2 ^a^	No. of Freq < 2 ^b^			
Overall (N = 181)	29/59	86/122	3.05 (1.51, 6.31)	2.82 (1.32, 6.18)	
**Cough**					0.943
Cough during day or night (N = 32)	6/17	8/15	3.46 (0.63, 23.92)	4.73 (0.51, 79.64)	
No self-reported cough (N = 149)	23/42	78/107	2.65 (1.18, 6.04)	2.90 (1.24, 6.98)	
**Age**					0.373
<35 years (N = 108)	11/34	47/74	3.52 (1.36, 9.65)	3.74 (1.25, 12.29)	
≥35 years (N = 73)	18/25	39/48	1.49 (0.42, 5.19)	1.35 (0.34, 5.12)	
**Body mass index**					0.078
<25 kg/m^2^ (N = 139)	20/45	68/94	3.56 (1.60, 8.18)	3.88 (1.65, 9.54)	
≥25 kg/m^2^ (N = 30)	7/11	12/19	0.32 (0.02, 2.88)	0.24 (0.01, 3.98)	

^a^. Numerator is the number of individuals who reported a house cleansing frequency of less than twice per week in the case group with denominator indicating the number of total cases in strata. ^b^. Numerator is the number of individuals who reported a house cleansing frequency of less than twice per week in the control group with denominator indicating the number of total controls in strata; ^c^. odds ratio adjusted for age, BMI and cough frequency; ^d^. odds ratio adjusted for age, BMI, cough frequency, physical activity, smoking, drinking, household income and highest education; ^e^. *p* value for the interaction term in mediation analysis.

## Data Availability

The data used and/or analyzed during the current study are available from the corresponding author on request.

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
