# Peer review of "Association between Low House Cleaning Frequency, Cough and Risk of Miscarriage: A Case Control Study in China"

_ijerph, 2021, doi:10.3390/ijerph18105304_

Round 1

Reviewer 1 Report

The Authors introduced several changes according to my suggestions as reviewer of previous submission. I do not have any further comments

Author Response

Thanks for your comments. I have no response.

Reviewer 2 Report

In my opinion the disproportions between two groups (case- 59 pregnant women& control-122 women who chose to conduct induced abortion) are too large .

In conclusions probably It would be better if you wrote : Home of pregnant woman should be cleaned at least twice per week. In that version you don't suggest that pregnant women should clean by themselves. Its important becasue in pregnancy they should avoid extense phisical effort. Moreover you do not pay attention on the products which are used during cleaning. The use of cleaning sprays, air fresheners and solvents during pregnancy may increase the risk of wheezing and infections in the offspring. Casas, L., Zock, J.P., Carsin, A.E. et al. The use of household cleaning products during pregnancy and lower respiratory tract infections and wheezing during early life. Int J Public Health 58, 757–764 (2013). https://doi.org/10.1007/s00038-012-0417-2

Orianne Dumas & Nicole Le Moual (2020) Damaging effects of household cleaning products on the lungs, Expert Review of Respiratory Medicine, 14:1, 1-4, DOI: 10.1080/17476348.2020.1689123

Author Response

Dear Reviewer:

We have received your helpful comments on the paper entitled:

Association between low house cleaning frequency, cough and risk of miscarriage: A case control study in China

Thank you very much for your valuable and constructive suggestions on this paper. We have carefully revised the manuscript according to your suggestions. The followings are our responses to your comments:

Comment 1:

In my opinion the disproportions between two groups (case- 59 pregnant women& control-122 women who chose to conduct induced abortion) are too large .

Response/Action

Thanks for your comment. Exactly, the disproportions between two groups (case- 59 pregnant women& control-122 women who chose to conduct induced abortion) are definitely a little large. This may be a limitation of this article. Clinically, the incidence of spontaneous abortion is 12-15%, so the collection of cases is relatively slow. We will further increase the number of cases during the subsequent studies.

Comment 2: 

In conclusions probably It would be better if you wrote : Home of pregnant woman should be cleaned at least twice per week. In that version you don't suggest that pregnant women should clean by themselves.  

Response/Action

Thanks for your advise suggestion. We have changed the description to “Home of pregnant woman should be cleaned at least twice per week.” in lines 24-25 and lines 269 in the revised manuscript.

This manuscript is a resubmission of an earlier submission. The following is a list of the peer review reports and author responses from that submission.

Round 1

Reviewer 1 Report

This study investigated the association between house cleaning frequency and the risk of miscarriage in a case control sample of Chinese population.
The interpretation of the results is not clearly presented and neither adequately supported by the evidence adduced.
Conclusions do not refers to the results of research.
I do not understand the message of this research.
The article doesn't possess scientific, practical value.

Author Response

Dear Reviewer:

We have received your helpful comments on the paper entitled:

Association between low house cleaning frequency, cough and risk of miscarriage: A case control study in China

Thank you very much for your valuable and constructive suggestions on this paper. I think I didn’t explain the article clearly, which caused a lot of misunderstandings. We have carefully revised and reorganize the manuscript . The followings are our responses:

  • The aim of the study was to investigate the association between environmental disruptor chemicals, life style exposure and female reproductive outcomes. This large study included exposure assessment in vitro and in vivo. The exposure assessment in vitro was performed by face-to-face questionnaire involved information regarding demographics-age, body mass index (BMI), current smoking or drinking status, household income, house cleaning frequency, etal. The clinical outcome was determined as miscarriage due to its increasing incidence. The vitro exposure by questionnaire provided that household dust maybe a potential clue for the occurrence of miscarriage, so we will furthermore collect the household dust from the two groups to determine the levels of environmental disruptor chemicals.We add the sentence in the line 37-41 in the revised manuscript:

“It has been suggested that exposure to common indoor environmental allergens could induce the occurrence of asthma, the main clinical manifestation of which is cough. Maternal asthma is associated with an increased risk of a spontaneous abortion due to the dysfunctional immune response. ”

  • In the results section, we made three analysis. Firstly, we compared the demographic characteristics of case and control patients and found that the case patients had more symptom of cough and less house cleaning frequency, compared to the control group (Table 1). Then, the inter-relationship test for house cleaning frequency, cough and risk for miscarriage was performed though adjusting different confounders (Table 2). It was interestingly showed that house cleaning of less than twice per week was significantly associated with cough during day or night, cough during day or night was significantly associated with risk of miscarriage, and house cleaning of less than twice per week was statistically significantly associated with miscarriage. Furthermore, the stratified analyses demonstrated that the association between house cleaning frequency and miscarriage was significant in participants without symptom of cough, participants < 35 years old and participants with BMI < 25 kg/m2 (Table 3).

We revised the first paragraph of the discussion lines 172-187 as follows:

“In the present study, we explored the association between lifestyle and miscarriage. The lifestyle questionnaire included age, body mass index (BMI), smoking, drinking, cough, physical activity, household income education and occupation. We found that in the miscarriage (case) groups, patients showed more symptom of cough and lower house cleaning frequency, compared to the patients of control. Then we further analyse the inter-relationship during the three elements. It was found house cleaning of less than twice per week was significantly associated with cough during day or night (OR=2.97, 95% CI: 1.36 ~ 6.75, P = 0.007), and cough during day or night was significantly associated with risk of miscarriage with OR of 2.69 (95% CI: 1.22 ~ 6.02, P = 0.014). Moreover, it was found that house cleaning of less than twice per week was statistically significantly associated with miscarriage with a high OR of 3.05 (95% CI: 1.51 ~ 6.31, P = 0.002). Otherwise, we verify the association of house cleaning frequency and miscarriage using various models and showed a increased risk of miscarriage in low house cleaning frequency patients in the different models. Moreover, in the stratified analyses, this trend was obvious in participants without symptom of cough, participants < 35 years old, and participants with BMI < 25 kg/m2. ”

  • In the discussion section, we increase the results of the stratified analyses in lines 233-240:

“In the stratified analyses, we found that the association of lower house cleaning frequency and high rish of miscarriage in participants without symptom of cough, participants < 35 years old, and participants with BMI < 25 kg/m2. This may due to the low statistical power caused by low sample size in these three groups (N = 35, N = 89, N = 39, respectively). In addition, the house cleaning may be just a potiental and weak influencing factor for the risk of miscarriage. In patients of advanced age, obesity and the syptom of cough, the effect of house cleaning would be not obvious. This is why the people don’t pay much attention to the habit of house cleaning.”

Reviewer 2 Report

Thank you for opportunity to review the presented paper.

I found it very interesting as  approaching so far  rather weakly elaborated hypothesis about association of between lifestyle, especially the house cleaning frequency, and miscarriage.

The strongest part of the paper is the discussion of results with special attention made o possible immunological connections between dust exposure and miscarriages.

On other hand the manuscript has some aspects which need corrections or at least some clarification,

  • The selection of controls – no information is provided how this was done. What was the source population and what was the criteria to select them. Maybe they were more often then cases employed and were so concentrated on their career that the less often cleaned their apartments. Or maybe they could afford some help to do it? These are just few speculations related to the possible bias created by selection of
  • Design of the study. As I read the “settings and sample” , it was clear to me that the Authors applied case-control approach. They recruited 59 pregnant women with clinical pregnancy loss as cases and 122 women who chose to conduct induced abortion as controls. However in table 1  cross-sectional approach was applied to present the data ( exposed vs less exposed women). This is not correct in my opinion as the cases and controls came from different population.  
  • Aim of the study – Authors made a statement that  not only explore the association between life- style, especially the house cleaning frequency, and miscarriage, but also “provide  the practical and feasible tips of staying at home for the women who try to conceive”. I have not found this in the paper.

Author Response

Dear Reviewer:

We have received your helpful comments on the paper entitled:

Association between low house cleaning frequency, cough and risk of miscarriage: A case control study in China

Thank you very much for your valuable and constructive suggestions on this paper. We have carefully revised the manuscript according to your suggestions. The followings are our responses to your comments:

Comment 1:

The selection of controls–no information is provided how this was done. What was the source population and what was the criteria to select them. Maybe they were more often then cases employed and were so concentrated on their career that the less often cleaned their apartments. Or maybe they could afford some help to do it? These are just few speculations related to the possible bias created by selection of

Response/Action

Thanks for your comment. The control and case population are from the same population in the outpatient during the same time. And the criteria of control population was the patient having a fetal bud body with a heartbeat during the trans-vaginal ultrasound examination, who want to conduct induced abortion. We revised the sentence “Meanwhile, patients having a fetal bud body with a heartbeat, who want to conduct induced abortion, were invited to participate in this study as controls. ” to “During the same time, patients having a fetal bud body with a heartbeat under the trans-vaginal ultrasound examination, who want to conduct induced abortion, were invited to participate in this study as controls.” in line 66-68 in the revised manuscript.

We supplement the sentence of “ The control and case population are from the same population in the outpatient during the same time.” in the 69-70 lines in the revised manuscript.

Furthermore we analyzed the occupation information from the questionnaire in the case and control groups and fond that the occupational distribution between the two groups were similar which we have supplemented in Table 1. Also there were no significant difference about the household income and highest education between the two groups. These information may indicate the consistency of the two groups of patients to a certain extent.

Comment 2: 

Design of the study. As I read the “settings and sample” , it was clear to me that the Authors applied case-control approach. They recruited 59 pregnant women with clinical pregnancy loss as cases and 122 women who chose to conduct induced abortion as controls. However in table 1  cross-sectional approach was applied to present the data ( exposed vs less exposed women). This is not correct in my opinion as the cases and controls came from different population.  

Response/Action

Thanks for your advise suggestion. These are two different research method. In fact, the case and control groups are from the same population and I have revised the whole Table 1 according to the case control approach.

We revised the description of the results in 125-133 lines in the revised manuscript, as follows:

“In total, there were 59 marriage patients (case) and 122 induced abortion patients (control). Mean age for them were 33.34±5.26 and 32.34±6.68, respectively. Mean BMI for them were 21.84±2.88 and 21.00±3.02, respectively. There were no significant differences between case and control groups in terms of smoking status, drinking status, household income, highest education and occupation (Table 1). In the case populations 17 reported a usual cough during day or night (42 no symptom of cough), while 14 among the 122 control patients, showing a significant p value. In addition, it was also demonstrated a lower house cleaning frequency (less than twice per week) in the case group compared to the control patients (p=0.005). ”

Comment 3:

Aim of the study – Authors made a statement that not only explore the association between life- style, especially the house cleaning frequency, and miscarriage, but also “provide  the practical and feasible tips of staying at home for the women who try to conceive”. I have not found this in the paper.

Response/Action

Thanks for your comment. The response to the statement of “provide  the practical and feasible tips of staying at home for the women who try to conceive” was the description of “Thus in order to reduce the risk of miscarriage, keeping a good cleaning up habit (more than twice per week) was a simple and feasible intervention for women who have conceived.” in lines 230-232 in the revised manuscript and the last sentence of the conclusion “Therefore, pregnant females are suggested to clean their home at least twice per week in order to avoid miscarriage.” in lines 262-264.

Reviewer 3 Report

This is an interesting study that provides proxy data on the effect of indoor air pollutants on the risk of miscarriage. The article is clear, well written.

The introduction should explain how cough could be a confounding or modifying factor in the relationship between the frequency of cleaning the house and miscarriages. The physiopathological mechanisms are explicit in discussion but the introduction in fact does not mention at all the taking into account of the cough which afterwards surprises when we see it appearing in the method part. Is cough on the causal path to miscarriage? I wonder to what extent it should not be studied separately as another explanatory factor for the risk of miscarriage. I would have liked to have the authors' opinion on this! It is not a confounding factor for me as such (like age for example or BMI).

There are a few points that need to be clarified. It appears that this study was not only set up to investigate the association between the frequency of home cleaning and the risk of miscarriage. As a result, it is unclear whether the authors conducted multiple analyses and found only this positive association that they present or whether this study is constructed as a sub-study of a larger study. It seems to me that this is important to clarify because it can considerably increase the risk of error if we are in the first case.

I disagree with the sentence “participants with cough showed a trend for higher risk for miscarriage compared with participants without cough although not statistically significant (OR = 4.73, 95% CI: 0.51 – 79.64)”. In view of the confidence interval, I don't think we can say anything about this result. Moreover, if I understand this painting correctly, this is not what it means. Does this table adequately represent the subgroup analyses for the association between exposure to low cleaning frequency and the risk of miscarriage? Which would mean, then, if you repeat your sentence, that among participants who cough, there is a greater risk of miscarriage for participants who do not clean their homes frequently compared to those who clean frequently?

In the method or discussion, the recruitment of controls should be discussed and justified. Are they from the same source population as the cases? Are the women who come to the center for abortion no different from the women who consult for miscarriages?

Author Response

Dear Reviewer:

We have received your helpful comments on the paper entitled:

Association between low house cleaning frequency, cough and risk of miscarriage: A case control study in China

Thank you very much for your valuable and constructive suggestions on this paper. We have carefully revised the manuscript according to your suggestions. The followings are our responses to your comments:

Comment 1:

The introduction should explain how cough could be a confounding or modifying factor in the relationship between the frequency of cleaning the house and miscarriages. 

Response/Action

In the introduction section, the sentence of “It has been suggested that exposure to common indoor environmental allergens could induce the occurrence of asthma, the main clinical manifestation of which is cough. Maternal asthma is associated with an increased risk of a spontaneous abortion due to the dysfunctional immune response.” was added to the lines 37-41 in the revised manuscript.

Comment 2:

Is cough on the causal path to miscarriage? I wonder to what extent it should not be studied separately as another explanatory factor for the risk of miscarriage. I would have liked to have the authors' opinion on this! It is not a confounding factor for me as such (like age for example or BMI).

Response/Action

Thanks for your comment. We assumed that cough is a mediator for association between house cleaning and risk of miscarriage. We analyzed the association between house cleaning and risk of miscarriage, between house cleaning and cough, between house cleaning, cough as independents and risk of miscarriage as dependent. In Table 2 model 2 and model 3, we showed that cough was not significantly associated with risk of miscarriage when it was adjusted in the model with house cleaning. Thus, we do not have enough statistical power to conclude whether cough is a mediator. However, the lower boundary for CI of cough was close to 1. Our results just suggests that cough might be a mediator, but need to be explored in a larger sample size in the future. In addition, in Table 3, we also explored whether cough is a moderator for association between house cleaning and risk of miscarriage. But the P value for the interaction term (cough*house cleaning) was not significant.

Comment 3:

It appears that this study was not only set up to investigate the association between the frequency of home cleaning and the risk of miscarriage. As a result, it is unclear whether the authors conducted multiple analyses and found only this positive association that they present or whether this study is constructed as a sub-study of a larger study. It seems to me that this is important to clarify because it can considerably increase the risk of error if we are in the first case.

Response/Action

In fact, a large study has been designed to investigate the association between environmental disruptor chemicals, life style exposure and female reproductive outcomes. This large study included exposure assessment in vitro and in vivo. The exposure assessment in vitro was performed by face-to-face questionnaire involved life style, eating habits and mental state. In addition, we collected the urine and blood samples to detect the level of exposure in vivo. Thus, this study was a sub-study of the large study. The results indicate that household dust maybe a clue for the occurrence of miscarriage, so we will furthermore collect the household dust from the two groups to determine the levels of environmental disruptor chemicals.

Comment 4:

I disagree with the sentence “participants with cough showed a trend for higher risk for miscarriage compared with participants without cough although not statistically significant (OR = 4.73, 95% CI: 0.51 – 79.64)”. In view of the confidence interval, I don't think we can say anything about this result. Moreover, if I understand this painting correctly, this is not what it means. Does this table adequately represent the subgroup analyses for the association between exposure to low cleaning frequency and the risk of miscarriage? Which would mean, then, if you repeat your sentence, that among participants who cough, there is a greater risk of miscarriage for participants who do not clean their homes frequently compared to those who clean frequently?

Response/Action

Thanks for your comment. We agree with you. In table 3, we in fact did stratified analyses. The sentence means that the association between house cleaning frequency and risk of miscarriage was stronger in people with cough compared to people without cough. However, considering that the CI was relatively large for people with cough, which is caused by its sample size and insufficient power, we cannot conclude anything. In addition, we reported the P value for interaction term (moderation analysis) in Table 3, and the P value for cough*cleansing was not significant. Therefore, we revised this sentence line 161-163 from the manuscript since we cannot reach to any robust point with the current data as follows:

In the mediation analyses, we were unable to show any significant moderation effect of cough, age or BMI on the association between house cleansing frequency and risk of miscarriage.

Comment 5:

In the method or discussion, the recruitment of controls should be discussed and justified. Are they from the same source population as the cases? Are the women who come to the center for abortion no different from the women who consult for miscarriages?

Response/Action

Thanks for your comment. The control and case population are from the same population in the outpatient during the same time. And the criteria of control population was the patient having a fetal bud body with a heartbeat during the trans-vaginal ultrasound examination, who want to conduct induced abortion. We revised the sentence “Meanwhile, patients having a fetal bud body with a heartbeat, who want to conduct induced abortion, were invited to participate in this study as controls. ” to “During the same time, patients having a fetal bud body with a heartbeat under the trans-vaginal ultrasound examination, who want to conduct induced abortion, were invited to participate in this study as controls.” in line 66-68 in the revised manuscript.

We supplement the sentence of “ The control and case population are from the same population in the outpatient during the same time.” in the 69-70 lines in the revised manuscript.

In our reproductive center, the clinical process of miscarriage patients and induced abortion patients was similar. They were all firstly diagnosed by trans-vaginal ultrasound examination. Then we will provide a more detailed consult. The miscarriage patients may be provided more examination such as embryo chromosome examination. The induced abortion patients were provided choice of suitable contraceptive methods.

Round 2

Reviewer 3 Report

The authors have responded to the comments made in the first review.